# SIGNS IN THE LOTTERY: STRUCTURAL SIMILARITIES BETWEEN WINNING TICKETS

## ABSTRACT

Winning tickets are sparse subnetworks of a deep network that can be trained in isolation to the same performance as the full network. Winning tickets have been found in many different contexts, however, their structural characteristics are not well understood. We propose that the signs of the connections in winning tickets play a crucial role. We back this claim by introducing a sign-based structural comparison metric that allows distinguishing winning tickets from other sparse networks. We further analyze typical (signed) patterns in convolutional kernels of winning tickets and find structures that resemble patterns found in trained networks.

## 1 INTRODUCTION

The lottery ticket hypothesis (Frankle & Carbin, 2019) claims the existence of sparse trainable subnetworks in a given deep network, the so-called winning tickets. While it has been known for a long time that artificial neural networks can be pruned significantly (removing more than 90% of the parameters) *after training* without impacting their performance (Liang et al., 2021; Blalock et al., 2020), such sparse networks usually resist training when their weights are initialized randomly, meaning they need longer to converge and usually reach lower performance. One interpretation of winning tickets is that they form the trainable core of a dense network, with the other parameters essentially being dead weights. This overparameterization of the dense network is still useful, as it combinatorially expands the number of subnetworks and hence the chance of containing a wining ticket.

Winning tickets have been reliably found for different types of deep networks by pruning connections from randomly initialized dense networks. However, the known pruning approaches are quite laborious, often requiring more resources than simply training the original dense network. A good characterization of what distinguishes winning tickets from other sparse networks still seems to be missing. Better understanding these properties could not only help in designing more efficient pruning algorithms, but it might also allow devising new initialization schemes for deep networks, leading to smaller networks and more efficient training.

In this paper, we address the question of whether winning tickets show specific structural characteristics that allow us to distinguish them from other sparse networks. We claim that it is not merely their sparse structure but also the sign of the connections that should be considered. After briefly reviewing previous work and motivating our approach (section 2), we introduce a sign-aware structural distance metric for sparse networks (section 3), explain our experimental setup (section 4) and apply our metric to the generated winning tickets. We then complement this quantitative analysis with a more qualitative inspection of spatial structures found in winning tickets (section 6) and conclude by summarizing our findings (section 7).

## 2 RELATED WORK

Since their discovery by Frankle & Carbin (2019), winning tickets have attracted a lot of attention and several works have shown their existence for different types of networks and datasets, although for larger networks, some additional warmup training seems to be required (Frankle et al., 2019). On the other hand, Ramanujan et al. (2020) have shown that there exist subnetworks that already

perform significantly above chance level without any training (specifically they extracted a subnetwork of an untrained ResNet-50 that performs like trained ResNet-34), and Malach et al. (2020) show, based on statistical arguments, that such networks should always exist. Frankle et al. (2021) claim that there are networks that do not only show good initial performance but can also be trained. Chen et al. (2021) propose a method to transform a wining ticket found in one network to be applied to another network architecture suggesting an underlying winning structure and Movva et al. (2021) analyze the structural overlap of winning tickets obtained from the same initial network.

Closest related to our study is the work by Zhou et al. (2019), who have conducted several experiments to analyze properties of winning tickets. These also include experiments investigating the role of connection signs. They tried different approaches to change the connection weights of a given wining ticket, including sampling weights from the original initialization distribution, reshuffling the connection weights, or assigning a constant value to all connections. While such a reinitialization typically destroys the winning property of a ticket, they observed that if the signs of the connections are kept, the resulting network still shows comparable performance to the original wining ticket. It is this observation that motivates us to take the sign into account when comparing the network structure.

## 3 STRUCTURAL COMPARISON OF SPARSE NETWORKS

In contrast to dense networks, sparse networks open a way to structural analysis based on their connection graph. For example, one can use methods from graph theory to compare the structure of two sparse networks $\mathcal{N}_1$ and $\mathcal{N}_2$. One such approach is the Neural Network Sparse Topology Distance (NNSTD) by Liu et al. (2020). This metric operates in a layer-wise fashion, considering for each pair of units the sets of incoming connections $G_1$ and $G_2$. It uses the normalized edit distance (NED), a normalized version of the graph edit distance, that counts the number of edits (additions and removals of connections) required to transform one network into the other. To account for differences between two networks, the units in two corresponding layers can be permuted to minimize the overall edit distance. The distance between $\mathcal{N}_1$ and $\mathcal{N}_2$ can then be obtained by averaging over the layer distances.

While the NNSTD can be used to find structural similarities between sparse networks, the structure of a network cannot be the sole factor distinguishing a winning ticket from other sparse networks: reinitializing a winning ticket with random weights usually destroys its winning property while keeping the sign, a winning ticket tends to stay trainable (Zhou et al., 2019). Hence we propose to use a structural similarity metric that takes the sign of a connection into account.

To obtain such a sign-aware metric, we adapt NNSTD in the following way: instead of just considering the set $G$ of all incoming connections to a node, we split that set into two parts, $G^+$ and $G^-$ consisting of only the positive and negative connections, respectively. The sign-aware $\text{NED}^\pm$ is then defined as the arithmetic mean of the NED for the positive and negative parts:

$$\text{NED}^\pm (G_1, G_2) = \tfrac{1}{2} \left( \text{NED} \left( G_1^+, G_2^+ \right) + \text{NED} \left( G_1^-, G_2^- \right) \right)$$

This metric penalizes connections with different signs in $\mathcal{N}_1$ and $\mathcal{N}_2$ more than a connection that is simply missing in the other network. One may choose to put more emphasis on a mismatch in positive or negative connections (by taking a weighted average; indeed in our experiment we observe, that the sign are usually not equally represented in the winning tickets). However, for this paper we stick to the symmetric definition.

Convolutional layers require a special treatment: such layers are considered to consist of multiple filters, each being a stack of 2-dimension kernels that are moved over the input tensor. We consider each filter as a unit, setting the set of incoming connections to be the entries in the filter that have not been pruned. The $\text{NED}^\pm$ is then computed for each pair of filters from the corresponding layers of $\mathcal{N}_1$ and $\mathcal{N}_2$ to obtain the $\text{NNSTD}^\pm$ for that layer.

## 4 EXPERIMENTAL SETUP

In designing our experiments, we follow the setting in the original paper by Frankle & Carbin (2019), using the same Conv-2 architecture, consisting of two convolutional layers with 64 kernels of size

Figure 1: The distribution of the sign-agnostic NNSTD (left) and sign-aware NNSTD$^\pm$ (right) computed distances for each condition, grouped by their category. The comparison indicates that taking the signs into account results in a clearer distinction between winning tickets and randomly pruned networks.[1]

$3 \times 3$, two dense layers with 256 units and an output layer with 10 units. We run our experiments on three different datasets: CIFAR-10, CINIC-10, and SVHN, all being scaled to the same size ($32 \times 32$ pixels, RGB) and labeled with 10 class labels, so that exactly the same network architecture can be used for all experiments.

We employ iterative magnitude pruning (IMP) with resetting strategy to obtain winning tickets. We run IMP for $N = 11$ rounds, in each round training for 10 epochs and then removing all connections not belonging to the top magnitude weights, to obtain the desired pruning rate after the final round. We use different pruning rates for the weights of the convolutional (68.6%, i.e., 10% per round), dense (91.4%, i.e., 20% per round) and output layers (68.6%), while not pruning the biases. Using this approach, we could reliably obtain winning tickets for each of the three datasets at a total pruning rate of 91.2%. For each dataset, we run 20 different trials, each time starting with a different random initialization. For comparison, we also create sparse networks by random pruning.

## 5 Structural Comparison of winning tickets

To find characteristic structures of winning tickets, we conduct several comparisons based on the NNSTD and our extended NNSTD$^\pm$ metric. In the *within-dataset WT-WT comparison*, we compute the similarity of two different winning tickets found for the same dataset. With $N = 20$ tickets, we have $N(N-1)/2 = 190$ possible comparisons from which 50 were chosen at random. As a baseline, we perform the same comparison also for the $N = 20$ random tickets (sampled from the same initial weights as the winning tickets). We then compare winning tickets to random tickets in the *mixed condition*. Finally, in the *between_DS1_DS2* condition, winning tickets from different datasets are compared.

---

[1]Note that the scales in this plot are not identical and a direct quantitative comparison of the two metrics is not intended. As NNSTD is not distinguishing between positive and negative weights it returns lower distances for all conditions. What is of interest, is the relative distance between the different groups in each plot.

Table 1: The distribution of distances computed by NNSTD and NNSTD$^{\pm}$ for each condition is statistically compared to the "control" condition *random*. Columns show the *absolute difference* of both means, the p-value of a *Welch Two Sample t-test* between both groups and a *Cohen's d* as the effect size. In contrast to the sign-agnostic NNSTD metric, the sign-aware NNSTD$^{\pm}$ metric consistently shows a huge effect size when comparing winning tickets to winning tickets within and also between datasets.

| Condition | NNSTD | | | NNSTD$^{\pm}$ | | |
|---|---|---|---|---|---|---|
| | difference mean | p-value | effect size | difference mean | p-value | effect size |
| WTs_CIFAR | -0.0006 | 0.307 | -0.27 (small) | -0.0122 | 2.5e-25 | -4.79 (**huge**) |
| WTs_CINIC | -0.0016 | 0.016 | -0.64 (medium) | -0.0102 | 2.1e-23 | -4.27 (**huge**) |
| WTs_SVHN | -0.0084 | 1.5e-12 | -2.49 (**huge**) | -0.0255 | 3.8e-32 | -7.94 (**huge**) |
| mixed_CIFAR | 0.0013 | 0.026 | 0.59 (medium) | 0.001 | 0.066 | 0.48 (small) |
| mixed_CINIC | 0.0017 | 0.003 | 0.8 (large) | 0.0015 | 0.006 | 0.74 (medium) |
| mixed_SVHN | 0.0051 | 6.3e-10 | 1.93 (very large) | 0.0035 | 1.3e-08 | 1.72 (very large) |
| CINIC_CIFAR | -3e-04 | 0.570 | -0.15 (negligible) | -0.01 | 1.9e-21 | -3.95 (**huge**) |
| SVHN_CIFAR | 0.0019 | 0.009 | 0.7 (medium) | -0.0096 | 9.4e-22 | -3.95 (**huge**) |
| SVHN_CINIC | 0.0013 | 0.078 | 0.46 (small) | -0.0067 | 2.3e-14 | -2.64 (**huge**) |

Figure 1 shows an overview of all comparisons applying the sign-agnostic NNSTD and the sign-aware NNSTD$^{\pm}$ metrics: the 10 conditions are grouped into the categories *WTs*, *mixed*, *random* and *between_DS*. The *WTs* category comprises all conditions, where a winning ticket from one dataset is compared to another winning ticket from the same dataset. In the *mixed* category, a winning ticket is compared to a random ticket. In *random*, two random tickets are compared. The category *between_DS* includes all comparisons between a winning ticket from one dataset, to a winning ticket from a different dataset.

In general, it seems as if NNSTD$^{\pm}$ is showing a greater similarity (a significantly lower distance) between WTs from the same or different datasets compared to the *random* condition than NNSTD. For the following analyses, the *random* condition will be our control group, to which the other conditions are statistically compared.

Table 1 compares the NNSTD-distances for the nine conditions to the *random* condition. The first column lists the differences in means. These differences seem very small throughout the conditions, but relative to the (very low) standard deviations of all groups, they can still be significant. To check this, a Welch Two Sample t-test was conducted, comparing the mean of each condition to the *random* condition and giving evidence of whether they are likely to originate from the same distribution. The Welch test was chosen, as the conditions were all approximately normally distributed, but did not necessarily have the same standard deviation. The null hypothesis is that both compared groups originate from a distribution with the same mean ($N_0 : \mu_{\text{test\_condition}} = \mu_{\text{random}}$), hence a two-tailed test was conducted. A significance value of $\alpha = 0.01$ was fixed, as comparing a lot of groups would lead to a higher probability of reaching the significance level in at least some groups, a smaller significance value seemed appropriate. The last column lists a Cohen's d to indicate the effect size. The effect size is calculated as: $d = \frac{\overline{x_c} - \overline{x_r}}{s}$ with s being the pooled standard deviation and $\overline{x_c}$ and $\overline{x_r}$ being the means of the currently relevant and the *random* condition. It therewith represents how many standard deviations the means of the two groups are apart. The effect size counts as: *negligible* $< 0.2 \leq$ *small* $< 0.5 \leq$ *medium* $< 0.8 \leq$ *large* $< 1.2 \leq$ *very large* $< 2 \leq$ *huge*.

It can be seen that, when comparing the conditions with the sign-agnostic NNSTD, the *mixed* conditions (at least two out of three) have a significantly higher distance than the *random* one. In the *WTs* conditions, only the *WTs_SVHN* has a significantly lower distance with a large effect size. From the *between* conditions, both conditions including SVHN-WTs have a higher distance than the control and the between CIFAR and CINIC group, although only significant in the *between_SVHN_CIFAR* condition.[2]

---

[2]More detailed plots and statistical analyses of each layer separately for NNSTD can be found in appendix A.1.

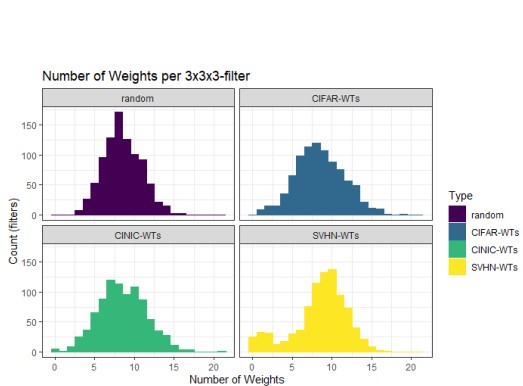



(a) Histogram of number of weights left per $3 \times 3 \times 3$ filter (from 0 to 27) for each group of subnetworks. Winning tickets show more sparse and more dense filters compared to random tickets.

(b) Visualization of how many weights are left in each position of the $3 \times 3 \times$ filters for each group of subnetworks (each in one row).

Figure 2: The distribution of remaining weights per filter for random tickets and winning tickets for the three datasets (CIFAR-10, CINIC-10, SVHN)

We next apply the sign-aware NNSTD$^{\pm}$-metric with subnetworks (WTs and random tickets) from the same pool as for the NNSTD-measure above. However, when computing the distances with NNSTD$^{\pm}$, 8 of the 9 "treatment" conditions have a significant difference in means to the "control" (*random*) condition, with a large effect size in most of these cases. Even more relevant, the different categories exhibit clear trends in the same direction. The *mixed* conditions have a positive effect size of around 0.5 to 1.7, meaning that the distances between a WT and a random ticket are (on average) about 0.5 to 1.7 standard deviations higher than in the *random* condition. All conditions comparing a WT to a WT (if within the same dataset or between different ones) exhibit a huge negative effect size, meaning that they have a lower distance (higher similarity) than random-random or WT-random (*mixed*) comparisons, making them structurally more similar to each other, according to NNSTD$^{\pm}$.

These results suggest, that (a) there are structural similarities between winning tickets, making them distinguishable from random tickets by applying the sign-aware NNSTD$^{\pm}$-metric, and (b) that these similarities at least partly depend on the signs or the connections, as they are not present in the sign-unaware NNSTD-comparison.

## 6  TYPICAL SIGN STRUCTURES IN WINNING TICKETS

The NNSTD$^{\pm}$ measurements suggest that WTs share similarities in their signed connection structure. This section analyzes these similarities in more detail, considering the winning tickets from all three datasets as well as the random tickets as a baseline. We focus on the first (convolutional) layer, as it exhibits very similar statistical results as the overall distances (the mean of all layers). Importantly, it shows a significantly lower distance for all WT conditions in comparison to the *random* condition.[3] In the following, we analyze which structural components are (more) present in the WTs (from all datasets) than in the random tickets.

The first aspect is the count of unpruned weights per $3 \times 3 \times 3$-filter. We observe that the winning tickets have a higher standard deviation in their count of remaining weights per filter than the random tickets[4], meaning that winning tickets rather contain filters with many remaining weights or with only few or no remaining weights, instead of pruning all filters to the same percentage. Histograms with the distribution of unpruned weights per filter can be seen in figure 2a.

---

[3]Exact numbers can be found in appendix A.1.

[4]While the random subnetworks have a standard deviation of 2.35 (on average, a filter has 2.35 weights more or less than the mean of 8.5), the winning tickets have a higher standard deviation of 3.05 (CIFAR-WTs), 3.03 (CINIC-WTs) and 3.45 (SVHN-WTs).

Thereupon we analyzed the count of unpruned weights in all positions of the $3 \times 3 \times 3$-filter. While the random subnetworks have a similar amount of remaining weights in each position, there are higher differences between positions in the winning tickets, as can be seen in figure 2b[5]. Higher and lower numbers of remaining weights (brighter and darker spots) suggest higher importance of weights in some positions compared to others. A special structure in the winning tickets is not evident though and these deviations might be dataset-dependant.

As a next aspect, we consider the ratio of signs of the weights. If randomly pruned, we could expect the ratio of the number of positive weights divided by the number of negative weights to be 1 (same amount of both signs). The random tickets have indeed a ratio of 1.01. The other groups of subnetworks have ratios of 1.83 (CIFAR-WTs), 1.57 (CINIC-WTs) and 2.02 (SVHN-WTs) though, having 1.5 to 2 times as many positive weights remaining as negative ones.

Following these three general aspects, we now combine positions and signs and study the patterns that can be found in the pruned kernels. For this, we will focus on the $3 \times 3$-kernels, not distinguishing between different colour channels.

Before considering specific patterns, we look at the frequency of combinations of weights. Figure 3 shows a map for each group of subnetworks and for each position in the $3 \times 3$ kernel, indicating how often another positive weight is jointly present in the same kernel. The red field shows the reference position for each plot.

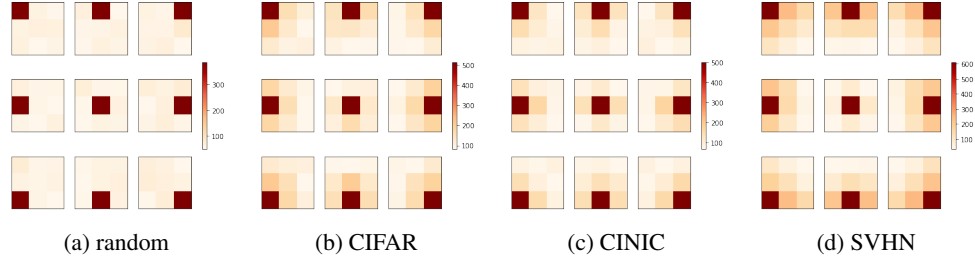

(a) random       (b) CIFAR       (c) CINIC       (d) SVHN

Figure 3: Heatmaps indicating how many positive weights at each position survived pruning when the weight at the reference position (the red field) is also positive.

In the random group, there do not seem to be more or less frequent combinations of unpruned weights. For the winning tickets this is not the case though: the positions close to the reference point more often also contain a positive weight than positions further away. When taking into account all kernels containing at least two positive weights and calculating the percentages of two positive weights being neighbours, the results are 34.5% for the random group, 45.9% for CIFAR-, 46.8% for CINIC- and 50.5% for SVHN-WTs. When doing the same computations with only negative values, the percentages are similar. The pruned kernels from the winning tickets seem to have "grouped" weights of the same signs left.

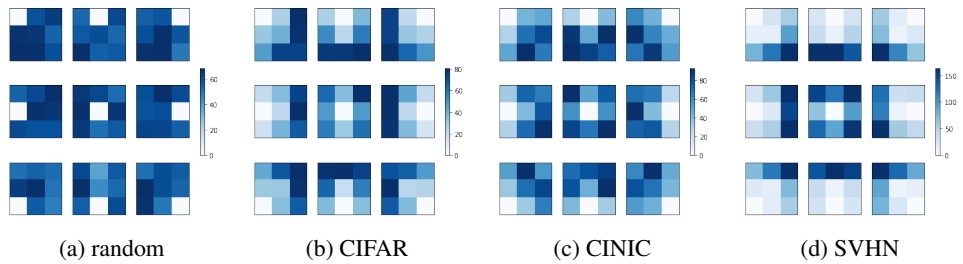

(a) random       (b) CIFAR       (c) CINIC       (d) SVHN

Figure 4: Heatmaps indicating the number of negative weights at each position that survived pruning when the weight at the reference position (the white field) is positive.

---

[5]This is also shown by the standard deviation (from the mean of 281 weights per position), which is 14.89 for the random group, 20.12 for the CIFAR-WTs, 24.84 for the CINIC-WTs and 54.11 for the SVHN-WTs.

The next interesting question is whether negative and positive weights also somehow influence the position of the oppositely-signed weights. Figure 4 shows the number of combinations of a positive weight in reference position with a negative weight in any other position. As can be seen, the negative weights rather occur further away from a positive weight when looking at the winning ticket groups.

After analyzing general combinations, we now turn to the actual patterns. Figure 5 shows which patterns of three unpruned weights are especially often encountered in the different subnetwork groups. The first nine most frequent patterns of each group are shown, red stands for a positive weight, blue for a negative weight. The same plot for four remaining weights can be seen in figure 6. For the other counts of weights left from 2 to 7, the corresponding plots can be found in appendix A.3.

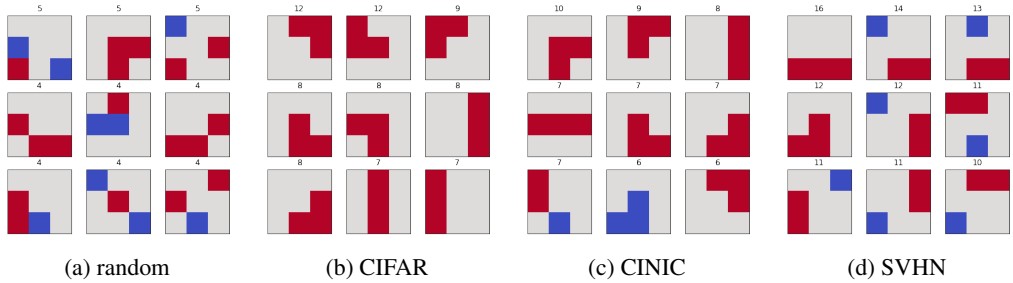

    (a) random          (b) CIFAR          (c) CINIC          (d) SVHN

Figure 5: The most frequently encountered patterns with three weights left in a 3x3 kernel. The number of occurrences is printed above the pattern. Red stands for positive weights, blue for negative ones.

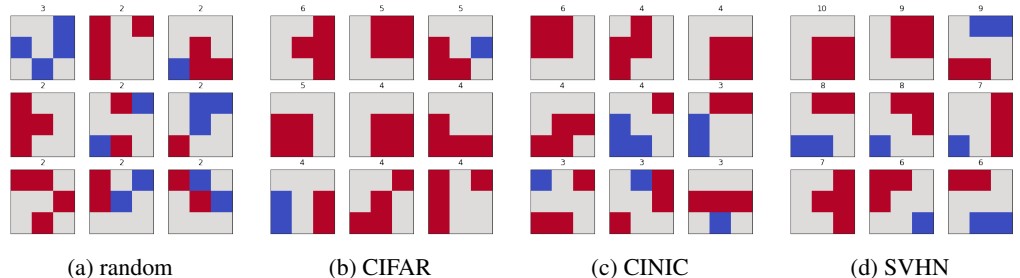

    (a) random          (b) CIFAR          (c) CINIC          (d) SVHN

Figure 6: The most frequently encountered patterns with four weights left in a 3x3 kernel. The number of occurrences is printed above the pattern. Red stands for positive weights, blue for negative ones.

As was already seen in the previous analysis of combinations of weights, remaining weights of the same sign seem to be rather close to each other and opposing signs rather further away. In general, one can observe that the most frequent patterns of the winning tickets do not seem random. Most kernels exhibit similarities to classical filters used in computer vision to detect an edge, a corner or a surface.

# 7 CONCLUSION

We have analyzed the signed structure of winning tickets obtained by the iterative magnitude pruning algorithm before training. We introduced a sign-aware metric NNSTD$^{\pm}$ to compare sparse networks, by extending the NNSTD metric. Using NNSTD$^{\pm}$ we were able to show that winning tickets are structurally more similar to each other than to random tickets, and also more similar than two random tickets compared with each other. This suggests the distribution of signs in a winning ticket to be a characteristic property. Interestingly, this is not only the case within, but also between different (image) datasets, suggesting that the structural aspects that make winning tickets special are (to some extent) dataset-independent.

We have further taken a look at the spatial arrangement of positive and negative weights in the first-layer convolutional kernels of winning tickets. These show characteristic patterns, with identically signed weights close to each other, resembling typical filters applied in computer vision.

These patterns would not be visible when ignoring the signs in the pruned kernels. Furthermore, these characteristic patterns are in clear contrast to randomly pruned kernels, indicating that these structures are advantageous for quick training, probably because they already anticipate the target configuration.

Both findings underline the significance of taking signs into account when analyzing trainable substructures of dense networks. While the topological structure alone seems not to be sufficient to explain winning tickets, considering also the initial signs changes the picture. Adopting this idea could not only lead to more efficient algorithms for finding winning tickets in dense networks, or creating sparse "trainable" networks from scratch, utilizing advantageous sign structures. In the long run, it may help to explain what makes these networks trainable and to design weight assignment procedures that go beyond random initialization by choosing signs in a structured way, allowing for smaller models and faster training.

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

## A  APPENDIX

The appendix provides additional figures and tables, adding some details to the figures and tables presented in the main text.

### A.1  SIMILARITY MEASURES PER LAYER (NNSTD$^\pm$)

Figure 1 in the main text shows the accumulated distances computed by NNSTD and NNSTD$^\pm$, which result from averaging over the five layers of the network. Figure A.1 shows the distances per layer for the NNSTD$^\pm$ metric:

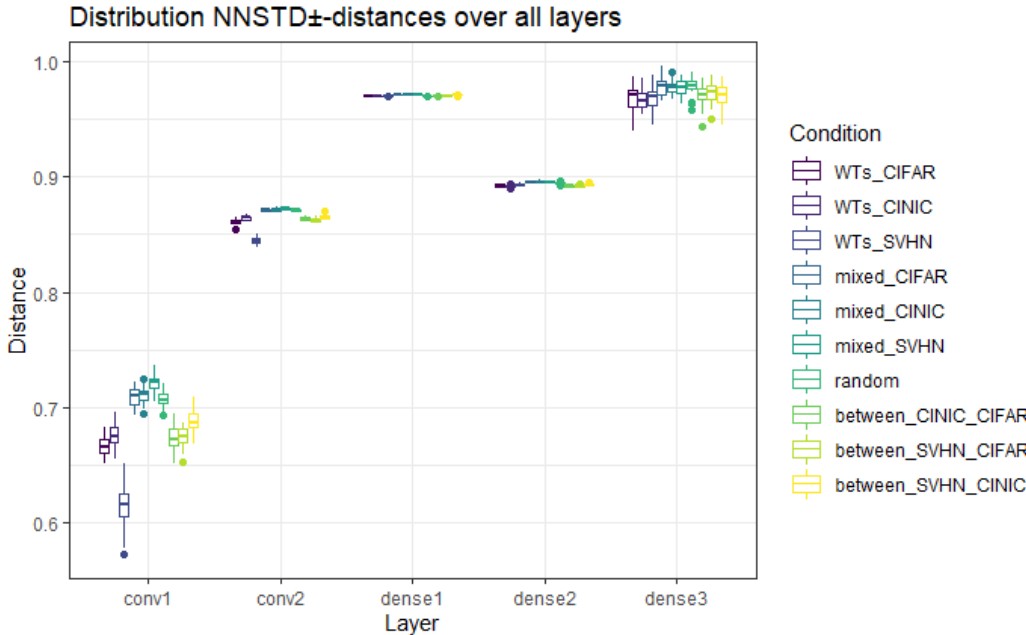

Figure A.1: The distribution of the NNSTD$^\pm$-computed distances over all layers for each condition.

The first thing one notices is that there is a huge difference between the distances of different layers, e.g. the first layers seem to be more similar to each other, than the later ones. These differences are due to the definition of the NNSTD$^\pm$, the different number of weights per layer and the different pruning rates of each layer. However, we are primarily interested in the differences between all conditions in the same layer, which are shown in mor detail in figure A.2.

The 5 layers are called *conv1* and *conv2* for the first and second convolutional layer, *dense1* and *dense2* for the next dense layers and *output* for the last dense output layer. Interestingly, all layers exhibit a similar trend (*mixed* has a higher or similar distance, *WTs* and *between_DS* a lower distance than *random*), except for the first dense layer, where every condition has a significantly higher distance than the *random* condition.

Table A.1 includes all statistical results for the individual layers computed with NNSTD$^\pm$. Layers *conv2*, *dense2* and *output* show a similar picture, all conditions comparing WTs (*WTs* and *between* conditions) show a very significant lower mean in distance than the *random* condition. The *mixed* conditions show varying results, only in the *dense2* layer, they all have significantly higher distances. The *dense1* layer shows a higher distance for *all* other conditions compared to the *random* one, with a huge effect size in each case, as it did with NNSTD.

### A.2  SIMILARITY MEASURES PER LAYER (NNSTD)

For comparison, we also document the similarity values computed by the unsigned NNSTD metric. Figures A.3 and A.4 show the distribution of the computed distances for each condition and

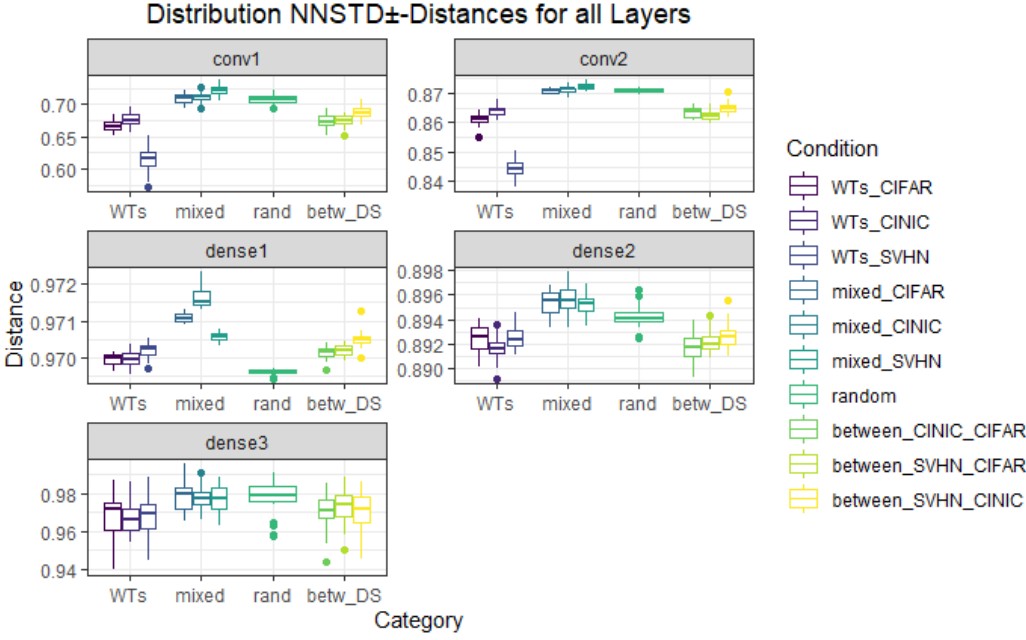

Figure A.2: The distribution of the NNSTD$^{\pm}$-distances for all layers separately

each layer separately. In the first plot, an overview of all layers is given. Again we can see a high difference between the distances of different layers, which appears to be quite similar to the same overview for NNSTD$^{\pm}$, shown in A.2. The second plot goes more into detail for each layer separately. Table A.2 shows the statistical results of comparing the distances of each condition to the *random* condition for each layer.

The first two convolutional layers, as well as the *dense2* layer, show similar results as the overall distances did. The *mixed* and *between* conditions show slightly higher distances, while the *WTs* conditions show varying results. Only the *WTs_SVHN* condition mostly exhibits a lower distance than the *random* condition. The *dense1* layer, similar to the NNSTD$^{\pm}$ results, shows a higher distance for *all* other conditions when compared to the *random* condition, with a huge effect size in every case. The *output* layer shows rather lower distances for all conditions compared to the *random* condition, but with varying significance and effect size. In this layer, WT comparisons (the *WTs* and *between* conditions) have a large effect size and significant differences in their mean of distances.

## A.3 FREQUENT PATTERNS IN WINNING TICKETS

Figures A.5, A.6, A.7, and A.8 show the most frequent kernels with 2, 5, 6, and 7 weights left, respectively, in the random and winning tickets with their corresponding absolute frequency. For each group, a total of $14 * 64 * 3 = 2,688$ kernels were considered. The total number of $3 \times 3$ kernels with $n$ (0-9) remaining weights can be found in table A.3, complementing the histogram in figure 2a in the main text.

Table A.1: The distribution of distances found by NNSTD$^{\pm}$ for each condition is compared to the "control" condition *random*. Columns show the *absolute difference in mean*, the p-value of a *Welch Two Sample t-test* between both groups and a *Cohen's d* as the effect size.

(a) Conv1

| Condition | diff. mean | p-value | effect size |
|---|---|---|---|
| WTs_CIFAR | -0.0395 | 4.6e-27 | -5.23 (huge) |
| WTs_CINIC | -0.0309 | 3.1e-20 | -3.82 (huge) |
| WTs_SVHN | -0.0898 | 9.9e-26 | -6.86 (huge) |
| mixed_CIFAR | 0.0027 | 0.151 | 0.38 (small) |
| mixed_CINIC | 0.0038 | 0.030 | 0.58 (medium) |
| mixed_SVHN | 0.0146 | 4.0e-11 | 2.1 (huge) |
| between_CINIC_CIFAR | -0.033 | 4.0e-20 | -3.88 (huge) |
| between_SVHN_CIFAR | -0.032 | 6.6e-23 | -4.26 (huge) |
| between_SVHN_CINIC | -0.0186 | 5.4e-13 | -2.43 (huge) |

(b) Conv2

| diff. mean | p-value | effect size |
|---|---|---|
| -0.01 | 1.1e-25 | -7.05 (huge) |
| -0.007 | 8.0e-22 | -5.28 (huge) |
| -0.0264 | 2.2e-32 | -13.12 (huge) |
| -0.0002 | 0.335 | -0.25 (small) |
| 0.0004 | 0.108 | 0.42 (small) |
| 0.0015 | 6.0e-09 | 1.8 (very large) |
| -0.0073 | 9.1e-23 | -5.6 (huge) |
| -0.0087 | 2.6e-26 | -6.91 (huge) |
| -0.0058 | 9.2e-20 | -4.55 (huge) |

(c) Dense1

| Condition | diff. mean | p-value | effect size |
|---|---|---|---|
| WTs_CIFAR | 0.0003 | 1.0e-13 | 2.82 (huge) |
| WTs_CINIC | 0.0004 | 1.9e-11 | 2.47 (huge) |
| WTs_SVHN | 0.0006 | 3.5e-18 | 4.13 (huge) |
| mixed_CIFAR | 0.0015 | 1.2e-45 | 15.05 (huge) |
| mixed_CINIC | 0.002 | 1.1e-27 | 9.33 (huge) |
| mixed_SVHN | 0.001 | 8.3e-38 | 9.97 (huge) |
| between_CINIC_CIFAR | 0.0005 | 4.1e-19 | 4.18 (huge) |
| between_SVHN_CIFAR | 0.0006 | 3.2e-25 | 5.63 (huge) |
| between_SVHN_CINIC | 0.0009 | 2.4e-22 | 5.7 (huge) |

(d) Dense2

| diff. mean | p-value | effect size |
|---|---|---|
| -0.0018 | 1.7e-08 | -1.7 (very large) |
| -0.0026 | 1.0e-15 | -2.82 (huge) |
| -0.0017 | 4.7e-09 | -1.78 (very large) |
| 0.0011 | 3.0e-05 | 1.17 (large) |
| 0.0013 | 3.4e-06 | 1.33 (very large) |
| 0.001 | 8.6e-05 | 1.09 (large) |
| -0.0025 | 3.3e-13 | -2.44 (huge) |
| -0.002 | 7.7e-12 | -2.21 (huge) |
| -0.0016 | 2.2e-08 | -1.68 (very large) |

(e) Output

| Condition | diff. mean | p-value | effect size |
|---|---|---|---|
| WTs_CIFAR | -0.0101 | 1.7e-04 | -1.04 (large) |
| WTs_CINIC | -0.0112 | 1.9e-06 | -1.37 (very large) |
| WTs_SVHN | -0.0102 | 4.3e-05 | -1.15 (large) |
| mixed_CIFAR | 0.0001 | 0.978 | 0.01 (negligible) |
| mixed_CINIC | -0.0002 | 0.905 | -0.03 (negligible) |
| mixed_SVHN | -0.0007 | 0.735 | -0.09 (negligible) |
| between_CINIC_CIFAR | -0.0079 | 0.001 | -0.91 (large) |
| between_SVHN_CIFAR | -0.006 | 0.009 | -0.7 (medium) |
| between_SVHN_CINIC | -0.0083 | 0.001 | -0.89 (large) |

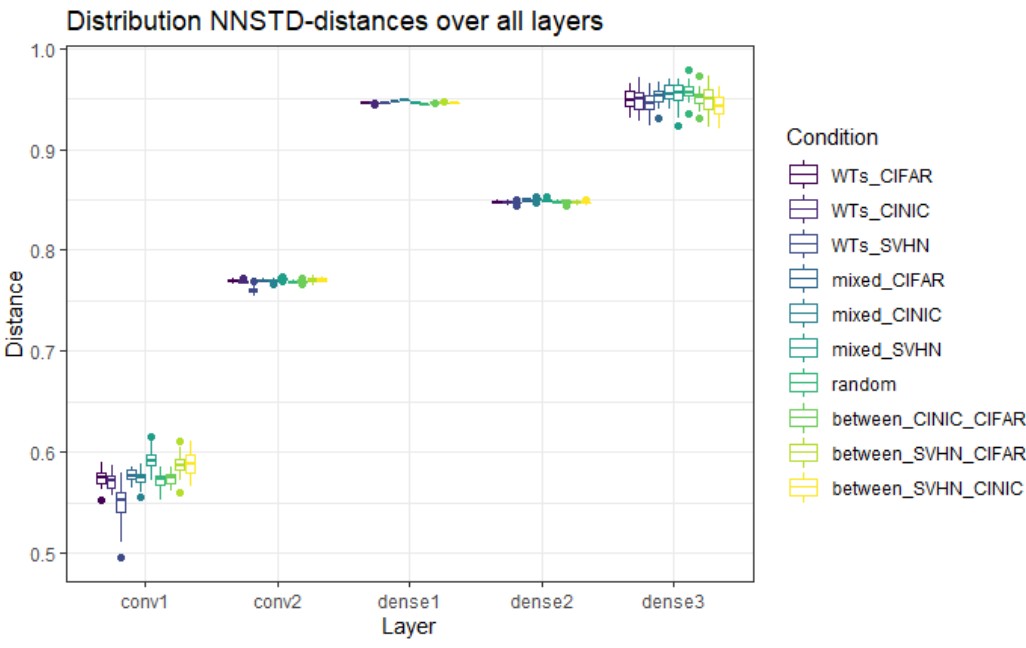

Figure A.3: The distribution of the NNSTD-computed distances over all layers for each condition.

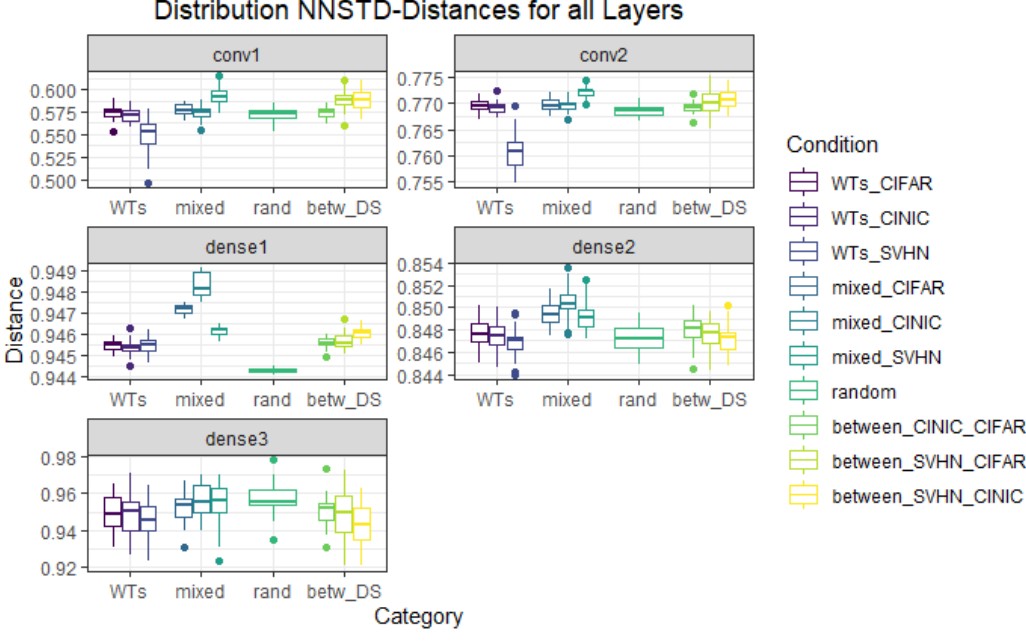

Figure A.4: The distribution of the NNSTD-computed distances for each layer separately

Table A.2: The distribution of distances found by NNSTD for each Condition is compared to the "control" condition *random*. Columns show the *absolute difference in mean*, the p-value of a *Welch Two Sample t-test* between both groups and a *Cohen's d* as the effect size.

(a) Conv1

| Condition | diff. mean | p-value | effect size |
|---|---|---|---|
| WTs_CIFAR | 0.0026 | 0.220 | 0.32 (small) |
| WTs_CINIC | -5e-04 | 0.823 | -0.06 (negligible) |
| WTs_SVHN | -0.0232 | 6.5e-07 | -1.54 (very large) |
| mixed_CIFAR | 0.0046 | 0.016 | 0.64 (medium) |
| mixed_CINIC | 0.0019 | 0.332 | 0.25 (small) |
| mixed_SVHN | 0.0206 | 6.9e-13 | 2.38 (huge) |
| between_CINIC_CIFAR | 0.0025 | 0.203 | 0.33 (small) |
| between_SVHN_CIFAR | 0.0159 | 2.6e-08 | 1.68 (very large) |
| between_SVHN_CINIC | 0.0157 | 2.8e-07 | 1.53 (very large) |

(b) Conv2

| diff. mean | p-value | effect size |
|---|---|---|
| 0.0011 | 0.001 | 0.92 (large) |
| 6e-04 | 0.050 | 0.52 (medium) |
| -0.008 | 3.0e-14 | -3.17 (huge) |
| 0.0012 | 3.9e-4 | 0.97 (large) |
| 0.001 | 0.001 | 0.88 (large) |
| 0.0036 | 4.5e-19 | 3.42 (huge) |
| 7e-04 | 0.016 | 0.64 (medium) |
| 0.0013 | 0.010 | 0.7 (medium) |
| 0.0022 | 9.9e-07 | 1.45 (very large) |

(c) Dense1

| Condition | diff. mean | p-value | effect size |
|---|---|---|---|
| WTs_CIFAR | 0.0012 | 2.6e-25 | 6.28 (huge) |
| WTs_CINIC | 0.001 | 8.2e-17 | 3.97 (huge) |
| WTs_SVHN | 0.0011 | 3.1e-15 | 3.61 (huge) |
| mixed_CIFAR | 0.0029 | 1.2e-41 | 16.17 (huge) |
| mixed_CINIC | 0.0041 | 3.5e-28 | 10.26 (huge) |
| mixed_SVHN | 0.0018 | 2.5e-32 | 9.79 (huge) |
| between_CINIC_CIFAR | 0.0013 | 3.8e-26 | 6.8 (huge) |
| between_SVHN_CIFAR | 0.0014 | 1.7e-18 | 4.62 (huge) |
| between_SVHN_CINIC | 0.0018 | 4.8e-27 | 7.81 (huge) |

(d) Dense2

| diff. mean | p-value | effect size |
|---|---|---|
| 0.0004 | 0.161 | 0.37 (small) |
| 0.0001 | 0.786 | 0.07 (negligible) |
| -0.0004 | 0.224 | -0.32 (small) |
| 0.0023 | 1.9e-09 | 1.84 (very large) |
| 0.0031 | 1.5e-12 | 2.33 (huge) |
| 0.0021 | 9.2e-08 | 1.58 (very large) |
| 0.0008 | 0.020 | 0.62 (medium) |
| 0.0004 | 0.255 | 0.3 (small) |
| 0 | 0.940 | 0.02 (negligible) |

(e) Output

| Condition | diff. mean | p-value | effect size |
|---|---|---|---|
| WTs_CIFAR | -0.0084 | 0.001 | -0.91 (large) |
| WTs_CINIC | -0.0094 | 4.7e-4 | -0.96 (large) |
| WTs_SVHN | -0.0113 | 9.8e-06 | -1.26 (very large) |
| mixed_CIFAR | -0.0046 | 0.031 | -0.57 (medium) |
| mixed_CINIC | -0.0014 | 0.540 | -0.16 (negligible) |
| mixed_SVHN | -0.0027 | 0.280 | -0.28 (small) |
| between_CINIC_CIFAR | -0.0069 | 0.002 | -0.84 (large) |
| between_SVHN_CIFAR | -0.0096 | 0.002 | -0.86 (large) |
| between_SVHN_CINIC | -0.0132 | 4.2e-06 | -1.33 (very large) |

Table A.3: The number of 3x3 kernels with *n* weights left for 14 pruned subnetworks selected from each group ($2,688$ kernels in total per group).

| group | 0 | 1 | 2 | 3 | 4 | 5 | 6 | 7 | 8 | 9 |
|---|---|---|---|---|---|---|---|---|---|---|
| random | 84. | 361. | 683. | 764. | 501. | 223. | 57. | 12. | 3. | 0. |
| CIFAR-WTs | 141. | 376. | 643. | 686. | 478. | 252. | 89. | 22. | 1. | 0. |
| CINIC-WTs | 149. | 367. | 632. | 689. | 489. | 264. | 76. | 18. | 4. | 0. |
| SVHN-WTs | 248. | 300. | 544. | 686. | 529. | 281. | 81. | 17. | 2. | 0. |

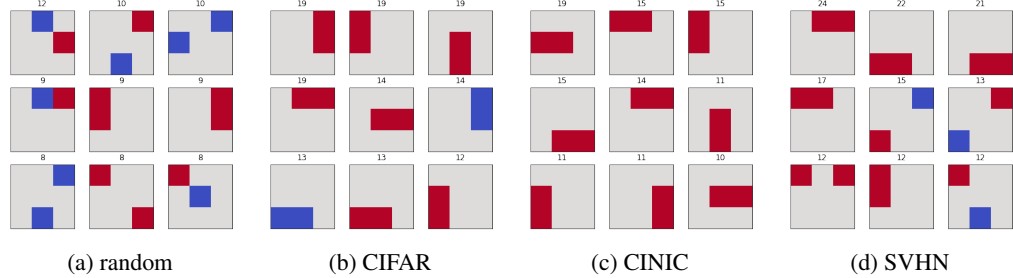

(a) random          (b) CIFAR          (c) CINIC          (d) SVHN

Figure A.5: The most frequently encountered patterns with two weights left in a 3x3 kernel. The number of occurrences is printed above the pattern. Red stands for positive weights, blue for negative ones.

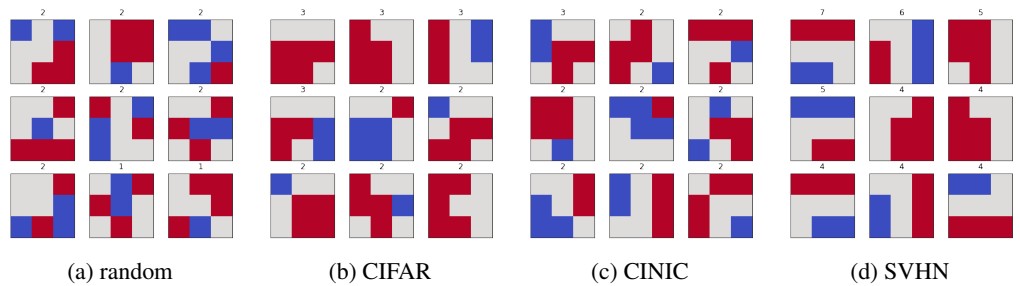

(a) random          (b) CIFAR          (c) CINIC          (d) SVHN

Figure A.6: The most frequently encountered patterns with five weights left in a 3x3 kernel. The number of occurrences is printed above the pattern. Red stands for positive weights, blue for negative ones.

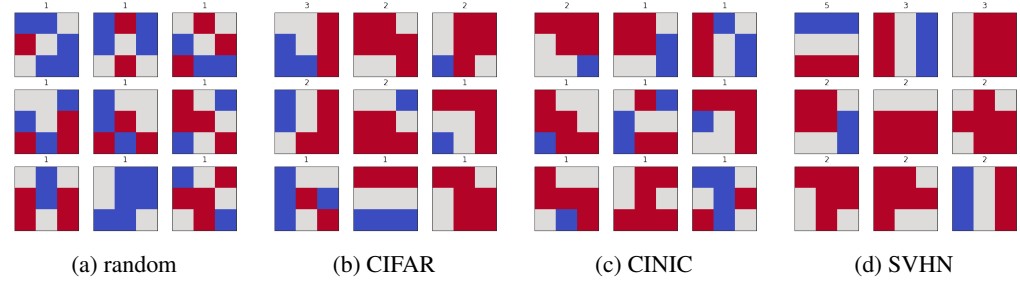

(a) random          (b) CIFAR          (c) CINIC          (d) SVHN

Figure A.7: The most frequently encountered patterns with six weights left in a 3x3 kernel. The number of occurrences is printed above the pattern. Red stands for positive weights, blue for negative ones.

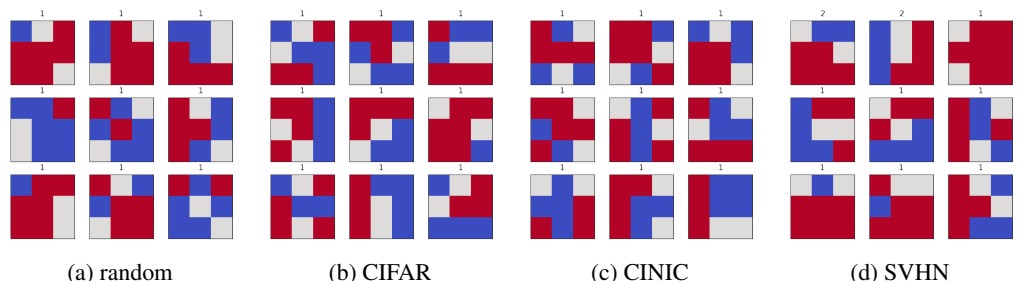

(a) random          (b) CIFAR          (c) CINIC          (d) SVHN

Figure A.8: The most frequently encountered patterns with seven weights left in a 3x3 kernel. The number of occurrences is printed above the pattern. Red stands for positive weights, blue for negative ones.

