# OpenReview forum: "Signs in the Lottery: Structural Similarities Between Winning Tickets"
_ICLR.cc/2023/Conference — Submitted to ICLR 2023_

### Official Review · Reviewer_nFr4 · 2022-10-22

**Confidence:** 4
**Correctness:** 3
**Technical Novelty And Significance:** 2
**Empirical Novelty And Significance:** 2
**Recommendation:** 3

**Clarity, Quality, Novelty And Reproducibility:**

The paper is okay and has every component to reproduce the work. However, the main argument needs to be supported with more experiments in various perspectives in a more aggressive way.

**Strength And Weaknesses:**

Pros

The main argument of the paper is that the winning tickets share similar network structures and the sign of the connections is an important characteristic. It is interesting to see the characteristics and it is good that the authors provided a simple measure that can evaluate this feature quantitatively.

Cons

However, the paper ends there. It only contains observations and there are few points that one can make out of the results. How can we benefit from knowing this? Can we somehow use this fact to find the winning ticket directly? Any insights? Any applications? Anything other than just observations?
In addition, the authors investigate the co-occurrence of signed weights in some specific convolution filters in the winning tickets and report that they seem to have some patterns. However, this is presented in a very abstract and qualitative way, which makes it hard to count on the results.
Overall, the main argument needs more rigorous backup experiments. For example, would the results still hold for different methods, like edge-popup? Different initialization distributions (Kaiming, signed, Gaussian, etc)? Different architectures? More results for different datasets? In my humble opinion, it is difficult to convince the reader in its current state.

The presentation can be improved. For example, it is very hard to see the differences in figure 1, which is one of the main results of the paper. Please make the colors stand out by using more vivid ones and consider using different markers or patterns than only using colors. The size of the fonts and tick values should be enlarged. (This also applies to other figures)


**Summary Of The Paper:**

This paper investigates a common structural characteristic between winning tickets. The authors argue that the found winning tickets indeed share a similar structure, the sign of the connections. To analyze this feature quantitatively, they devise a new metric (sign-aware structural distance) that measures the degree of the sign and structural similarity between winning tickets better (than solely comparing the network structure). Based on their analyses and experiments on different image datasets, sign-aware structural similarities between winning tickets seem to be a dataset-independent characteristic.

**Summary Of The Review:**

Overall, the idea of the paper is interesting, but further research is needed beyond simple observation.

---

### Official Review · Reviewer_ghvk · 2022-10-24

**Confidence:** 4
**Correctness:** 2
**Technical Novelty And Significance:** 2
**Empirical Novelty And Significance:** 1
**Recommendation:** 1

**Clarity, Quality, Novelty And Reproducibility:**

While this paper is, I believe, fundamentally flawed in its analysis/method, the writing itself is good. There are some relevant references (Zhou et al. in particular), but the background doesn't cite any of the papers that have improved understanding of Lottery Tickets since Zhou, and so is lacking substantially.

* The statement in the Abstract that "We propose that the signed of the connections in winning tickets play a crucial role" is confusing at best and misleading at worst in that the authors are not the first to propose this, indeed in the body of the paper it is motivated by the previous work of Zhou et al.


**Strength And Weaknesses:**

Strengths
* There is less analysis about the structure (i.e. mask) of winning tickets than the weights and this paper, at least at a high level, proposes better understanding the importance of structure of winning ticket subnetworks.
* The results in Table 1 showing differences between the metric distance between WTs and random tickets within the $NNSTD^{\pm}$ are potentially interesting.

Weaknesses
* It appears the paper compares Winning Tickets (i.e. subnetworks with masks found from pruning a dense neural network (i.e. the pruning mask), and using the (original) Lottery Ticket Hypothesis, initialized with the original pruned dense initialization, with sparse subnetworks - with *random masks* (i.e. as described in Section 4, "random pruning"). While they appear to still use the original pruned dense initialization, this doesn't make much sense, as we don't expect completely random connectivity to perform as well as that found via training in winning tickets! Rather (as the authors point out themselves in section 3) the main question in this space is why we can't train from *random initialization* the *same* subnet, i.e. *same pruning mask*, to as good a generalization as in a winning ticket.
* The paper only looks at winning tickets in the context of the original lottery ticket hypothesis - i.e. using dense initialization and tiny models. We now know in practice this work doesn't extend to larger models, and instead we must use weight-rewinding in practice. Similarly it's not clear that any analysis the authors make on these tiny models/original winning tickets would extend to real-world models.
* The proposed "metric" itself ($NNSTD^{\pm}$) is not directly comparable to vanilla NNSTD (expanded upon in the next weakness), as the authors themselves point out in the footnote on page 5, these are fundamentally different metrics. However, the author's claim that the "relative distances between the different groups in each plot" is comparable is also not necessarily true! These relative distances are still distances in different metric spaces. This is fine when the authors compare the separation of WT v.s. random within the same metric as done with Table 1.
* It's not clear that the proposed (different) $NNSTD^{\pm}$ metric for convolutional NNs is comparable to the one for fully-connected neural networks given the modifications suggested.

**Summary Of The Paper:**

The authors propose a sign-based metric for comparing the structure of two sparse subnetworks, in order to understand the importance of signs in Winning Tickets (Lottery Tickets). Using two different metrics, one that is "sign-aware", they compare the distances between Winning Tickets and "random tickets", which are in this particular work defined as random masks/random weights. The authors further attempt to analyze differences between signed patterns in sparse subnetworks found for convolutional neural networks.

**Summary Of The Review:**

While the direction of understanding the importance of the mask itself in sparse neural networks is interesting, the comparison of winning tickets with masks found from training to models with random masks is not well motivated. Furthermore, its not clear that the analysis presented in this paper is significant, and there are several reasons to question the comparison of models even within the same metric, as it differs for convolutional and fully-connected models.

---

### Official Review · Reviewer_UJuy · 2022-10-30

**Confidence:** 4
**Correctness:** 2
**Technical Novelty And Significance:** 3
**Empirical Novelty And Significance:** 3
**Recommendation:** 5

**Clarity, Quality, Novelty And Reproducibility:**

Clarity: While the paper is relatively clear and easy to follow, it would benefit by a careful proofreading and editing to follow a more rigorous and academic writing style as discussed above.

Quality: The paper seems to have a fair quality level, but some aspects of it need improvement (see above).

Originality: The paper has a fair level of novelty. The related work could be extended to cover better the topic of sparse neural networks.

Reproducibility: In my opinion, it is possible but not very easy to reproduce the work with the current level of details. A better methodology description together with, perhaps, the release of open-source code can increase the reproducibility level.




**Strength And Weaknesses:**

Strength:
* The topic addressed is well understudied even if a small progress on it could help in understanding better the performance and the behavior of sparse neural networks
* The proposed methodology has a fair level of novelty, in my opinion
* The empirical validation shows that the proposed method, NNSTD±, can achieve a better granularity and clarity than the baseline method (e.g., better clustering of the output like in Fig 1) in illustrating the differences between sparse neural networks models.

Weaknesses:
* The proposed method, NNSTD±, could benefit in clarity by a more thorough mathematical description and qualitative discussions in Section 3. For instance, a graphical representation of the method to improve its readability, adding more mathematical details on how the method actually works, adding an algorithm, the range of the metric output, etc.
* The empirical validation, even if it seems to be well executed, it is incomplete as it addresses just convolutional layers. A systematic study on other type of layers (e.g., fully-connected) could make stronger the results and bring more confidence in the metric.
* The related work is very poor, and this leads to several statements in the paper which are not fully accurate. While, up to my best knowledge, there is very little work in the literature addressing this particular problem (i.e., the distance between two sparse neural networks), there is a considerable amount of work on dense-to-sparse and sparse-to-sparse training. With respect to the statements, here are some examples: “…such sparse networks usually resist training when their weights are initialized randomly…”; “…However, the known pruning approaches are quite laborious, often requiring more resources than simply training the original dense network…”; “…Since their discovery by Frankle & Carbin (2019), winning tickets have attracted a lot of attention…”. It was actually shown from 2017 [1], that a sparse neural networks trained from scratch can easily outperform its equivalent dense counterparts (where the latter has exactly the same amount of neurons) by using the sparse evolutionary training algorithm which put the basis of what is called today dynamic sparsity, prune and grow strategies, or dynamic sparse training [2]. Non-exhaustively, starting from these two missing references a proper related work discussion can be made.
* Accordingly, the writing style can be improved to be more rigorous and more academic. All concepts have to be properly defined. E.g., the “winning ticket” term needs a proper definition somewhere in the introduction or in a background section.

References:

[1] Decebal Constantin Mocanu, Elena Mocanu, Peter Stone, Phuong H. Nguyen, Madeleine Gibescu, Antonio Liotta, Scalable Training of Artificial Neural Networks with Adaptive Sparse Connectivity inspired by Network Science, Nature Communications 2018, https://arxiv.org/abs/1707.04780

[2] Torsten Hoefler, Dan Alistarh, Tal Ben-Nun, Nikoli Dryden, Alexandra Peste, Sparsity in Deep Learning: Pruning and growth for efficient inference and training in neural networks, JMLR 2021, https://www.jmlr.org/papers/volume22/21-0366/21-0366.pdf



**Summary Of The Paper:**

The paper addresses the problem of understanding the differences between various sparsely connected neural networks models. Consequently, it proposes an improvement over the current existing method (metric) which measures the distance between two sparse neural network models, i.e. Neural Network Sparse Topology  Distance (NNSTD), by considering also the sign of the weight values. The new proposed method is named NNSTD± and it is evaluated mainly on convolutional neural networks.

**Summary Of The Review:**

The paper addresses a timely and relevant topic with a fair level of novelty. Unfortunately, given the above discussed limitations, I believe that the paper is not ready yet for publication in its current form.

---

### Official Review · Reviewer_MFkC · 2022-10-30

**Confidence:** 3
**Correctness:** 3
**Technical Novelty And Significance:** 2
**Empirical Novelty And Significance:** 3
**Recommendation:** 3

**Clarity, Quality, Novelty And Reproducibility:**

Not very clear. Need further improvements.

The method section is less than one page.

Also, the page seems not ready for review yet.

**Strength And Weaknesses:**

Strength:

* The topic of whether lottery tickets (LTs) is unique or share some similarity is of great interests to the communicty.

Weakness:

* The paper does not organized very well.
* What is the conclusion of this paper? The sign is only thing matters or is one part of it. LTs may have lots of similarity or dissimilarity. For example, [1] aruges that the ratio of remaining neurons in each layer matter the most. How is this paper different from that one?

[1] Sanity Checks for Lottery Tickets: Does Your Winning Ticket Really Win the Jackpot?

**Summary Of The Paper:**

This paper investigate the structrual similarity between winning tickets. In particular, they argue that the signs in weight connection play a critical role.

**Summary Of The Review:**

I have to reject this paper in the current shape. I can see the point in the results but overall, the writing need huge improvements.

The authors are welcomed to further revise this paper to be ready for review.

---

### Decision · Program_Chairs · 2023-01-20

**Decision:**

Reject

**Justification For Why Not Higher Score:**

N/a

**Justification For Why Not Lower Score:**

N/a

**Metareview: Summary, Strengths And Weaknesses:**

The authors suggest that the signs of the connections in winning tickets are significant, and propose a sign-based structural comparison metric to distinguish winning tickets from other sparse networks. They further analyze typical patterns in convolutional kernels of winning tickets and discover structures that resemble patterns found in trained networks. This research indicates that the signs of connections in winning tickets are a crucial factor in network performance.

The reviewers identified the following strengths and weaknesses about the paper

Strengths:
1. winning tickets share similar network structures and the sign of connections is an important characteristic.
2. A simple measure evaluating this feature.

Weaknesses:
1. The paper needs more rigorous backup experiments.
2. Results presented in a abstract and qualitative way, which makes it hard to count on the results.
3. The presentation can be improved.
4. There is perhaps little insight into how this fact can be applied or utilized.

Since all scores were reject, and the authors did not engage in the rebuttal, the paper is not recommended for publication.